# Hemodynamics and Tissue Optical Properties in Bimodal Infarctions Induced by Middle Cerebral Artery Occlusion

**DOI:** 10.3390/ijms231810318

**Published:** 2022-09-07

**Authors:** Chun-Wei Wu, Jia-Jin Chen, Chou-Ching K. Lin, Chien-An Chen, Chun-Ie Wu, Ing-Shiou Hwang, Tsung-Hsun Hsieh, Bor-Shing Lin, Chih-Wei Peng

**Affiliations:** 1School of Biomedical Engineering, College of Biomedical Engineering, Taipei Medical University, Taipei 11031, Taiwan; 2Department of Biomedical Engineering, College of Engineering, National Cheng Kung University, Tainan 70101, Taiwan; 3Department of Neurology, National Cheng Kung University Hospital, College of Medicine, National Cheng Kung University, Tainan 70101, Taiwan; 4Institute of Allied Health Sciences, College of Medicine, National Cheng Kung University, Tainan 70101, Taiwan; 5Department of Physical Therapy, College of Medicine, National Cheng Kung University, Tainan 70101, Taiwan; 6Department of Physical Therapy and Graduate Institute of Rehabilitation Science, College of Medicine, Chang Gung University, Taoyuan 33302, Taiwan; 7Neuroscience Research Center, Chang Gung Memorial Hospital, Linkou 33305, Taiwan; 8Department of Computer Science and Information Engineering, National Taipei University, New Taipei City 237303, Taiwan; 9School of Gerontology and Long-Term Care, College of Nursing, Taipei Medical University, Taipei 11031, Taiwan

**Keywords:** ischemic stroke, middle cerebral artery occlusion, near infrared spectroscopy, cerebral hemodynamics, interhemispheric correlation coefficient, cerebral blood flow, absorption, scattering

## Abstract

Various infarct sizes induced by middle cerebral artery occlusion (MCAO) generate inconsistent outcomes for stroke preclinical study. Monitoring cerebral hemodynamics may help to verify the outcome of MCAO. The aim of this study was to investigate the changes in brain tissue optical properties by frequency-domain near-infrared spectroscopy (FD-NIRS), and establish the relationship between cerebral hemodynamics and infarct variation in MCAO model. The rats were undergone transient MCAO using intraluminal filament. The optical properties and hemodynamics were measured by placing the FD-NIRS probes on the scalp of the head before, during, and at various time-courses after MCAO. Bimodal infarction severities were observed after the same 90-min MCAO condition. Significant decreases in concentrations of oxygenated hemoglobin ([HbO]) and total hemoglobin ([HbT]), tissue oxygenation saturation (StO_2_), absorption coefficient (μa) at 830 nm, and reduced scattering coefficient (μs’) at both 690 and 830 nm were detected during the occlusion in the severe infarction but not the mild one. Of note, the significant increases in [HbO], [HbT], StO_2_, and μa at both 690 and 830 nm were found on day 3; and increases in μs’ at both 690 and 830 nm were found on day 2 and day 3 after MCAO, respectively. The interhemispheric correlation coefficient (IHCC) was computed from low-frequency hemodynamic oscillation of both hemispheres. Lower IHCCs standing for interhemispheric desynchronizations were found in both mild and severe infarction during occlusion, and only in severe infarction after reperfusion. Our finding supports that sequential FD-NIRS parameters may associated with the severity of the infarction in MCAO model, and the consequent pathologies such as vascular dysfunction and brain edema. Further study is required to validate the potential use of FD-NIRS as a monitor for MCAO verification.

## 1. Introduction

Stroke is the leading cause of long-term adult disability and the second leading cause of death worldwide [1,2]. Considering that human stroke is extremely complex in its causes, expressions, and vasculature that is involved, a standardized animal model is needed to explore the pathophysiology and therapeutic approaches in a highly reproducible and well-controllable way [3]. The middle cerebral artery (MCA) is the largest cerebral artery that supplies most of the brain, and is also the most often affected by stroke: about half (50.8%) of all ischemic strokes occur in MCA and its branches [4,5]. For this reason, middle cerebral artery occlusion (MCAO) using intraluminal filament to generate infarcts in MCA territory has been developed [6,7]. The model involves introducing a nylon filament directly into the internal carotid artery (ICA) until it blocks the cerebral blood flow (CBF) to the MCA [3]. This method does not require craniectomy and allows permanent occlusion or temporary one followed by reperfusion. With these features plus clinical ischemia analogy, the MCAO by intraluminal filament has become one of the most widely used models in animal models of stroke [3,6,7,8,9,10,11,12]. Despite many studies leading the optimization of the methodological parameters [13], the intraluminal placement of the filament is still a surgical challenge. Since the improper position of the filament can cause insufficiency in MCA blood flow disruption and further prevent infarct formation [14,15], monitoring of CBF during the MCAO surgery has become an urgent need to check the success of the occlusion. CBF during the surgery is usually monitored using laser Doppler flowmetry (LDF) [16,17,18,19,20,21,22,23,24,25], less common using laser speckle contrast imaging (LSCI) [26,27] and magnetic resonance imaging (MRI) [15,22,28,29]. However, applying LDF or LSCI requires an incision of the scalp and thinning of the skull to allow the penetration of laser light into brain tissue. This procedure will cause tissue damage and increased surgical burden. MRI can be performed in a non-invasive way without damaging the scalp and skull, but it is not convenient for real-time monitoring of the surgery inside the MRI machine. These disadvantages limit the use of LDF, LSCI, and MRI during MCAO surgery. A better way to monitor the cerebral blood supply during MCAO for researchers to confirm the success of the modeling, and to explore the pathological changes after MCAO is still a significant demand.

Taking advantage of light in the near-infrared (NIR) range (600–900 nm), which can penetrate a few centimeters into tissues [30], near-infrared spectroscopy (NIRS) can non-invasively detect concentration changes of hemoglobin in the brain [31]. NIRS has the advantages of portability, non-radiation, high temporal resolution, and motion–artifact tolerability, thus having the potential of becoming widely applicable for the long-term monitoring of stroke and neurovascular diseases [32,33,34]. Three types of NIRS systems have been developed (Figure 1). Continuous-wave (CW-) NIRS can detect relative changes of the hemoglobin level based on that the absorption of the light is proportional to the amount of hemoglobin. Frequency-domain (FD-) NIRS uses light that is amplitude-modulated at radio frequencies (several hundred MHz) and measures the attenuation in amplitude and phase shifts of the transmitted signal to estimate absorption and scattering of the NIR light by tissue. Time-domain (TD-) NIRS detects the temporal distribution of photons produced when a short-duration (several picoseconds) laser pulse is transmitted through the tissue, and then estimates the absorption and scattering properties. With source–detector pairs separated by multiple distances, both TD- and FD-NIRS can provide absolute quantification of the hemoglobin level based on calculation derived from the absorption and scattering properties of the tissue and the molar extinction coefficient of hemoglobin [35,36]. The major parameters of absolute quantitative NIRS include the absorption coefficient (μa), reduced scattering coefficient (μs’), and concentrations of oxygenated ([HbO]) and deoxygenated hemoglobin ([HbR]) within the optical field. Total hemoglobin concentration ([HbT]) can be obtained by summing [HbO] and [HbR], and tissue oxygenation saturation (StO_2_) can be derived from the percent ratio between [HbO] and [HbT] [35].

The strengths of NIRS, including long-term continuous monitoring capability and high temporal resolution, make it a promising tool for measuring cerebral hemodynamic oscillation during stroke evolution. Previous studies used CW-NIRS to explore the temporal correlation between two hemodynamic oscillations obtained from both hemispheres in unilateral stroke patients [37,38,39]. The interhemispheric correlation coefficient (IHCC) was derived from computing the zero-lag cross-correlation of two hemodynamic oscillated signals from lesioned and non-lesioned hemispheres. These studies reported that stroke patients had lower IHCC when compared with normal subjects [37,38] or those with mild strokes [39]. These results suggested that the desynchronized hemodynamic oscillation between lesioned and non-lesioned hemispheres may be associated with the severity of the unilateral infarction. However, the relationship between IHCC and the progression of cerebral infarction requires further investigation.

NIRS has been utilized to investigate cerebral oxygenation and hemodynamics in rodent MCAO models [40,41,42,43,44,45]. Most of these studies used CW-NIRS to monitor the relative changes of hemoglobin during and after ischemia [40,42,45]; some used FD-NIRS to obtain the absolute values of cerebral oxygenation and tissue optical properties [41,43,44]. Meanwhile, all these studies performed on the MCAO model were focused on the transition of the ischemia only within several hours. A comprehensive study that includes all the NIRS parameters and covers extensive time courses to investigate sequential hemodynamic changes after MCAO is needed. Therefore, the aim of the study was to monitor the changes in tissue optical properties (μa and μs’) and cerebral hemodynamics ([HbO], [HbR], [HbT], StO2, IHCC) during and after MCAO using quantitative FD-NIRS. By investigating the temporal changes in FD-NIRS parameters, we expected to explore the feasibility of using FD-NIRS for MCAO verification.

## 2. Results

### 2.1. MCAO Lesion

Although consistent MCAO surgery was performed, TTC staining on the third day after MCAO showed two types of outcomes: (i) small-sized infarctions mainly in subcortical tissue, some involving the ventrolateral cortex, and (ii) large infarctions covering subcortical tissue and the ventrolateral and dorsolateral cortices (white signals in Figure 2A). These two types of infarction sizes were confirmed by clustering analysis using *k*-means algorithm. Based on the clustering, the outcomes of the infarction can be divided into the mild infarction group and the severe infarction group (Figure 2B). The percent of infarction was 8.08 ± 4.19% (mean ± SD) for mild infarction (*n* = 14) and 29.44 ± 2.23% for severe infarction (*n* = 14). The NIRS parameters derived from mild and severe groups were then further analyzed for statistical difference (see Section 2.2). The extent of brain edema induced by infarction was estimated by measuring the percent of volumetric change in the lesioned hemisphere. The percent values of swelling were 7.33 ± 4.62% (mean ± SD) for mild infarction and 14.30 ± 8.00% for severe infarction (Figure 2C). A significant difference was detected using a two-tailed unpaired *t*-test (*p* = 0.011).

### 2.2. Quantification of Optical and Hemodynamic Parameters in Lesioned Hemisphere

The absolute quantification of tissue optical properties such as μa and reduced μs’, and cerebral hemodynamics ([HbO], [HbR], [HbT], and StO_2_) before, during, and after MCAO surgery was performed using the multiple-distance FD-NIRS. Although there was some contribution from the circulation in the scalp, multi-distance montage (source–detector pairs at 3, 6, and 9 mm) allowed us to detect hemodynamic signals that most likely arose from cerebral tissues. The absolute values of μa and μs’ at two wavelengths (690 and 830 nm), as well as the values of [HbO], [HbR], [HbT], and StO_2_ measured from the lesion hemisphere at various time courses were summarized (Figure 3 and Table 1). The F values and P values derived from the two-way ANOVA were summarized (Table 2). The results suggest that the time course affects all NIRS parameters significantly (*p* < 0.05), whereas severity alone had no significant effect on any of the parameters. However, significant interactions between the time course and severity were shown for all NIRS parameters (*p* < 0.05), which indicates that severity does not have the same effect for all time courses (or that the time course does not have the same effect for all groups).

Tukey’s post-hoc tests were employed to further compare the respective means of all NIRS parameters between any two of the three groups (sham, mild, and severe infarction) at each time course (Figure 3). For the absorption coefficients, significant decreases were observed during occlusion (occ) and d1, while significant increases were detected on d3 for severe infarction. As for the scattering coefficients, significant decreases were detected during occ, with increases on d2 and d3, in severe infarction. Hemodynamic parameters were calculated from the absorption and scattering properties. During occ, significant decreases in [HbO], [HbT], and StO_2_ were observed in severe infarction. All hemodynamic values returned toward the baseline (pre) values when the filament was withdrawn 90 min after MCAO (post) through d2. Finally, the values of [HbO], [HbT], and StO_2_ derived from MCAO groups were observed to be higher than those derived from the sham group on d3.

### 2.3. IHCCs in MCAO

IHCCs derived from [HbO] oscillations from all groups at various time courses are plotted as means ± SEM (Figure 4). An IHCC of 1 indicates that the hemodynamic oscillations from the two hemispheres are perfectly symmetrical (in phase); IHCC values of 0 and −1 indicate 90° and 180° phase delays between hemispheres, respectively. After Fisher’s Z-transformation, two-way repeated-measures ANOVA showed significant main effects of time course (F_5,90_ = 14.69, *p* < 0.0001), severity (F_2,18_ = 19.10, *p* < 0.0001), and interaction (F_10,90_ = 2.24, *p* = 0.0219). Significant decreases were shown in IHCCs from both mild and severe infarctions during the occlusion (occ) by Tukey’s post-hoc test (versus sham). However, during the reperfusion (post), a significant decrease was observed between severe versus sham and severe versus mild infarction. There was no significant difference between the mild versus sham groups. No difference was found among groups on d1 to d3.

## 3. Discussion

### 3.1. Previous NIRS Studies of Cerebral Hemodynamics in MCAO Model

NIRS can non-invasively measure cerebral hemodynamics in cortical tissue through the intact scalp in a real-time manner. These features make NIRS a promising tool for MCAO surgery monitoring. Reduced partial oxygen pressure in the brain associated with MCAO has been reported [17]. Therefore, we hypothesized that cerebral hemodynamics derived from NIRS parameters (including tissue oxygenation and hemoglobin levels) may be associated with the outcome of ischemic injury in MCAO model.

Our results of cerebral hemodynamic changes in severe infarction are comparable with previous studies using NIRS to directly probe the lesioned hemisphere of MCAO animal. First, in the acute stage (pre, occ, and post) of the MCAO surgery, decreased [HbO], [HbT], and StO_2_ have been observed during the occlusion (occ). As the intraluminal filament was withdrawn after MCAO to allow reperfusion (post), [HbO], [HbT], [HbR], and StO2 values gradually returned toward the sham control values [40,41,42]. Next, three days after the surgery, significantly increased [HbO], StO_2_, and [HbT] were found in severe infarction compared to those from sham and mild infarctions. These findings are similar to those obtained from the permanent MCAO model [41]. Interestingly, the reduction of CBF obtained from the core MCA territory was compared with the cerebral tissue oxygenation derived from FD-NIRS. No significant correlation was reported [41].

Several imaging systems based on the principle of NIRS have been established for acquiring spatial information on the hemodynamic changes in the cerebral ischemic model [43,44,45,46]. A study used spatially modulated NIR illumination to reconstruct the two-dimensional (2-D) image of cerebral hemodynamics in the model of proximal MCAO [43]. Another reported the development of a novel diffuse optical tomography (DOT) by combining the spatial-distributed multi-channel FD-NIRS with diffuse correlation flowmetry. The system can obtain the 2-D image of hemoglobin concentrations and relative CBF to further reconstruct the dynamic imaging of StO_2_, relative cerebral blood volume, and cerebral metabolic rate of oxygen consumption during MCAO [44]. Furthermore, others applied the volumetric finite element model to generate 3-D DOT imaging from the embolic MCAO model with thrombolytic therapy [45] or common carotid artery occlusion (CCAO) models [46]. Due to the highly scattering property of the light propagating in tissue, the spatial resolution of the NIRS is usually a drawback. The developments of these topographic and tomographic methods compensated for the poor spatial resolution, and have demonstrated the utility of NIRS in providing a spatiotemporal profile of cerebral hemodynamics during ischemic events. DOT may be a promising tool to reveal the relationship between collateral circulation and MCAO variation in further investigation.

Changes in cerebral blood oxygenation during the MCAO model are compared with those obtained in the human ischemic stroke study. In the study of human ischemic stroke in MCA territory, the values of StO_2_ were significantly lower on the infarcted side compared to those on the non-infarcted side on the first-day post-stroke. The difference disappeared from the third to the fifteenth days, with the values of StO_2_ on the infarcted side gradually increasing throughout the monitoring period [47]. Our findings of cerebral blood oxygenation in the days after MCAO are similar to those obtained in a human study. However, the potential use of cerebral hemodynamics as biomarkers for clinical stroke still requires further clinical study to overcome the variation in NIRS measurements that is supposedly due to probe montage and anatomical variations of individuals.

### 3.2. MCAO Variation and LDF Monitoring

Using the same MCAO surgical procedure and occlusion duration to induce cerebral infarction in rat, various infarct sizes were observed in the current study, as ever reported previously [18,19,22,27,48,49,50]. Diverse severity of ischemic lesions similar to our current findings has been observed by MRI [22,48,49] and tissue staining [18,19,27,50]. Certain proportions of MCAO surgeries ending in mild infarction brought up the necessity of monitoring the CBF in the MCA territory during the surgery to prevent insufficient occlusion. Ipsilateral CBF monitoring using LDF has been demonstrated efficient to detect successful occlusion [16,17,18,19,20,21,22], or premature reperfusion and subarachnoid hemorrhage (SAH) [23,24,25]. The successful occlusion was induced when the insertion of the filament caused a significant drop in ipsilateral CBF of MCA territory to 20~30% of the baseline. In about 25% of the cases, premature reperfusion was identified as a steep rise in continuous ipsilateral LDF signal during the occlusion, and it may account for the variation in infarct size and neurological outcome associated with the MCAO model [23,24,51,52]. The detection of premature reperfusion during the occlusion allowed the investigators to readjust the placement of the filament so that the adequate MCAO can be accomplished [23,24].

It seems useful using LDF during MCAO to overcome insufficient occlusion; however, discrepancies were also reported regarding probing MCA territory with continuous ipsilateral LDF to monitor the decrease in CBF induced by MCAO [50,53,54]. The reduction of CBF from ipsilateral MCA territory did not match histological infarction [50,53]. The study suggested that early behavioral tests may predict early death and no infarction from successful infarction and SAH when all of these groups show decreased CBF (<30% of baseline) in MCA territory during occlusion [50]. The other two studies also reported LDF did not significantly affect the MCAO outcome [53,54].

Potential explanations for the discrepancies include collateral circulation [16,19,22] and anatomical variations of the vasculature [14,55], which are also possible factors for the variability of infarct size. A study discussed that measurements of CBF at the core of the MCA territory during MCAO usually showed very strong reductions, and a very narrow range of CBF values hardly matched up to a rather wide range of infarct sizes. Furthermore, they proved that infarct size is dependent on the reduction in collateral circulation, which can be monitored by LDF targeting perifocal MCA territory [16]. Later, a team used multi-site LDF to compare the CBF in the core MCA territory and the border zone between the cortical branches of the anterior cerebral artery (ACA) and MCA during MCAO. Their findings demonstrated that the reduction of CBF in the collateral territory is significantly correlated with neurological score and infarct size; while no correlation was observed in the case of the core MCA territory was probed [19,22]. Interestingly, they also reported similar bimodal infarct typologies as we did in the current study, and their monitoring of collateral flow with multi-site LDF agreed with acute MRI for the prediction of the MCAO bimodal outcomes.

Detecting collateral flow using multi-site LDF seems to be a sensitive method to predict MCAO outcomes; however, probing the collateral flow correctly can be tricky. A single LDF probe can only detect a very limited area and it is quite difficult to visually target the collateral circulation during the surgery, not to mention the anatomical variations of the vasculature from each animal and surgical bias from each laboratory, which make it even harder to practice. Using imaging technologies such as LSCI [26,27] and MRI [15,22,28,29] to obtain the spatial distribution of the CBF changes from core and collateral territories can be one of the alternative options. However, we just mentioned the limitation of using LSCI and MRI in MCAO monitoring in the introduction. In this study, we measured the expression of hemodynamic parameters other than CBF obtained from FD-NIRS during the MCAO. Our findings suggested the potential of using NIRS-derived hemodynamics for non-invasive MCAO monitoring.

### 3.3. Changes in Tissue Scattering in Severe MCAO

In the current study, the severe infarction group expressed decreased μs’ during MCAO, and increased μs’ on d2 and d3 after MCAO. These sequential patterns of μs’ are not like simultaneously recorded μa, hemoglobin levels, or StO2, which are highly relative to the changes in concentration of hemoglobin [32]. We assumed the decrease in μs’ during MCAO may be due to the changes in cellular conformation induced by acute injury, and the increase in μs’ on d2 and d3 may be associated with brain edema induced by ischemic injury hours to days later. To further interpret our assumptions on the observations in μs’, several studies with consistent or controversial findings are discussed in the following paragraphs.

Decreases in μs’, the scattering amplitude, and scattering power during MCAO compared to the pre-MCAO baseline have been reported [43]. It has been previously shown that decreases in scattering power were associated with increases in scatterer size [56]. Meanwhile, decreases in scattering amplitude reflected the changes in scatterer density, distribution, and refractive index of the tissue [57]. Therefore, the author supposed that their observation of decreased scattering during MCAO may be due to the change in cellular structure in response to acute injury [43]. Others investigated changes of NIR light scattering imaging in the model of permanent MCAO to monitor spatiotemporal distribution of spreading depolarization (SD) and development of the infarction [58]. In the core area of infarction, a sudden drop in scattering intensity induced by MCAO was detected and followed by a gradual increase and repetitive wavelike changes induced by SDs. In the peripheral region of the infarction, instead of a sudden drop immediately after the MCAO, a gradually increase followed by repetitive SDs were observed through the whole time. The authors speculate that the gradual increased scattering after MCAO may be associated with morphological alteration due to neuronal injury, although the discussion on the sudden drop of the scattering in the beginning of the MCAO was lacking [58]. We propose that our finding of decreased μs’ was consistent with previous reports in MCAO [43,58]. 

Delayed increases in μs’ shown on d2 and d3 led us to further discussion on brain edema induced by brain injury. Using NIRS to monitor changes in μs’ in the traumatic brain injury (TBI) model, an increase in μs’ was observed from the first hour through to five days after TBI, with a peak on the third day [59]. Furthermore, they found a strong positive correlation between values of μs’ and brain water content following TBI with or without dehydration treatments [59]. The relationship between changes in NIR light scattering and brain swelling induced by water intoxication with or without mannitol treatment was further investigated [60]. They found that both NIR light scattering and intracranial pressure (ICP) increased after water intoxication. The increased scattering found in in vivo brain and in vitro brain slices could both be reversed after dehydration treatment. Furthermore, they also demonstrated that the scattering intensity was highly correlated with brain water content (r = 0.98) [60]. 

We found significant brain swelling in the severe infarction on d3 compared to the mild infarction. Based on our current findings and previous studies [59,60], we hypothesized that the delayed increase in μs’ on d2 and d3 from the severe infarction was associated with brain edema. However, the time point is still controversial. Water content from the infarcted hemisphere significantly increased at 6 h after MCAO and lasted for 2 days, with a peak on the first day [61]. A similar edema phenomenon was confirmed by chronic ICP monitoring in MCAO rats for 5 days [62]. In our observation, no difference was detected in μs’ on d1 when comparing any two of the severe, mild, or sham groups. Therefore, the relationship between μs’ and brain edema induced by MCAO requires further study.

### 3.4. IHCC

Cerebral spontaneous hemodynamic oscillation in the low-frequency range was found to be associated with blood pressure fluctuation, and the myogenic, metabolic and neurogenic controls involved in maintaining the CBF constant during blood pressure fluctuation [63,64]. Featured as a high temporal resolution, NIRS has been demonstrated as a useful tool to study the alterations in cerebral hemodynamic oscillation in healthy subjects [63,65], stroke patients [37,38,39,64], and animal MCAO [42].

An animal study involves using NIRS to detect cerebral hemodynamic fluctuation induced by vasospasm in the model of MCAO [42]. The authors observed periodic fluctuations in changes in both [HbO] and [HbT] from the ischemic hemisphere during occlusion. Meanwhile, Fourier power spectral analysis showed that the power of the fluctuating signal from the rats without pre-treatment of the anti-vasospasm drug was significantly higher than those with pre-treatment. This study demonstrated the potential application of NIRS to monitor hemodynamic fluctuation associated with vascular dysfunction during MCAO [42]. 

IHCC was first introduced to detect a temporal correlation between hemodynamics from both hemispheres in ischemic stroke patients previously [37]. The authors obtained hemodynamic signals using CW-NIRS probing both MCA territories in normal and sub-acute stroke patients. The IHCCs were computed from hemodynamic signals in physiological high-frequency ranges, such as cardiac (0.7–3 Hz) and respiratory (0.15–0.7 Hz) ranges. Their results showed lower IHCCs expression in the stroke subjects compared to the healthy ones, which indicated interhemispheric desynchronization in strokes. They speculated that the reduced IHCCs might result from disrupted cerebral autoregulation [37]. Later, reduced IHCCs were further confirmed in stroke patients performing cycling tasks [38]. It seems that IHCCs derived from the premotor cortex and sensorimotor cortex might reveal the impairment that is associated with the brain responding to motor tasks. Recently, IHCCs in stroke patients with various severities were further investigated [39]. Unlike previously reported decreases in IHCCs in the high-frequency range (0.15–2 Hz) [37], the results showed no significant difference in the high-frequency range. However, in low- (0.06–0.15 Hz) and very low- (0.02–0.06 Hz) frequency ranges, significant differences were found in reduced IHCCs among various severities. Hemodynamic oscillations in low-frequency ranges reflected the activities of myogenic, metabolic, and neurogenic controls of the vasculature [63,64]. Therefore, the authors discussed the myogenic and neurogenic damages in stroke, and suggested the desynchronizations in low- and very low-frequency ranges are associated with these damages in different severity. Furthermore, a significant correlation was also found between IHCCs and neurological scores. In sum, these findings suggest the relationship between hemodynamic desynchronization and stroke severity [39]. 

In this study, we explored the temporal profile of hemodynamic desynchronization between both hemispheres from severe and mild infarction using IHCC. The 0.01–0.8 Hz frequency range, which is lower than the heart rate and the respiration rate of rats, was extracted from [HbO] traces for IHCC calculation. The results showed overall decreased IHCCs in both the mild and severe infarctions during occlusion. Of note, a further reduction was observed in the severe infarction after reperfusion (post), while a gradual recovery was seen in the mild infarction (Figure 3), then both severe and mild infarction were returned toward the baseline on d1 to d3.

Our IHCC result showed a remarkable difference after reperfusion when comparing severe MCAO with the mild one. We hypothesize that severe infarction may suffer more hypoxic damage during the occlusion versus the mild one (Figure 3H), and that the damage induces more severe vascular dysfunction, which was revealed by desynchronization in the low-frequency range after reperfusion. Such vascular dysfunction in stroke, which may be referred to as impairment in cerebral autoregulation [37] and neurovascular coupling [66], vasospasm [42], etc., requires further investigation.

### 3.5. Limitations of the Study

There are some limitations in the current study that need to be addressed. First, the types and conditions of the cerebral ischemic model were limited. There are so many factors and protocols affecting the outcome of cerebral ischemia, as we just mentioned in the introduction section. In the current study, we only applied proximal MCAO with 90-min occlusion by silicon-coated microfilament in isoflurane-anesthetized male SD rats following Koizumi’s method. The reproducibility and universality of monitoring cerebral ischemia by FD-NIRS require further studies.

Parameters that can affect the infarct variability and outcome reproducibility of the MCAO model usually include strains, age, body weight, body temperature, anesthesia, occlusion duration, type of intraluminal filament, and other surgical procedures [13]. Furthermore, the anatomical variations in cerebrovasculature associated with strains can affect the variation of infarction [13,14,55]. A meta-analysis compared the coefficient of variation of infarct size induced by various MCAO protocols in different rat strains; Wistar had significantly lower variability than Sprague–Dawley (SD) [13]. Variance in infarct size from SD rats may be due to the ICA branches that can support MCA territory, which were identified in 80% of the animals [14]. A review article then discussed that Wistar Kyoto (WKY) is a better choice than SD for stroke modeling because it lacks the vascular variability of the SD [55]. Theoretically, vascular variation shall affect NIRS result as well. Parallel studies using brain imaging to provide structural information and spatial hemodynamics are critical for validation.

Finally, there are some major drawbacks of NIRS including limited penetration depth, poor spatial resolution, and contamination by signals from the skin, muscle, and scalp [30,32,33,34]. Improvements are needed for probe montage so that the multiple-distance arrangement can target a sufficient depth of tissue to dilute the interference from the superficial signals. An efficient montage can also target regions of interest with less trade-off in spatial resolution. Although the motion tolerance of NIRS is relatively better than other brain images (such as MRI, LSCI, LDF, etc.), the slight displacement of a probe against the tissue surface during the measurement can still create significant motion artifacts. Therefore, a solid holder or interface is needed as well to mount the probes stably on the head during MCAO surgery in the supine position. In addition, featured as high temporal resolution, NIRS has an advantage in continuous monitoring in a real-time manner. With the help of a modified interface to smoothly transit from pre-surgery, during, and after surgery, a complete temporal profile shall be available.

### 3.6. Potential Clinical Applications

Continuous measuring of hemoglobin and tissue oxygenation level using NIRS may benefit acute stroke patients [32,33,34]. In the current study, we identified sequential changes in the brain hemodynamics and optical properties acquired with NIRS during and after rat MCAO. These findings may inspire the applications of NIRS for early diagnosis of ischemic stroke since NIRS is quite safe, affordable, and feasible for long-term use [67]. However, the translational potential of these parameters in human stroke requires further studies because of the complexity of the stroke onset, individual variation of the stroke patient, and reproducibility of the NIRS measurement.

The current finding is made from the ischemic stroke model, while the hemorrhagic stroke which contributes 20% of the total stroke is also critical [1,2]. Intracranial aneurysm (IA) is a risk for hemorrhagic stroke, and MCA has a high potential for developing an IA [68]. The relationship between changes in CBF and wall shear stress has been studied for the rupture risk using computer simulation [69,70]. Meanwhile, cerebral autoregulation has been shown to disrupt after aneurysmal subarachnoid hemorrhage [71,72]. Therefore, we speculate that IHCC, which is affected by disrupted cerebral autoregulation [37], may be a potential indicator for aneurysmal subarachnoid hemorrhage detection.

Non-invasive monitoring of the intracranial fluid content, such as intracranial compliance using MRI, has become an emerging need for clinical use in various types of hydrocephalus [73]. In the previous paragraph (Section 3.3), we discussed that the increased μs’ may be associated with brain edema [59,60]. Since the μs’ obtained by non-invasive NIRS is an absolute quantity, μs’ is comparable between individuals and may have the potential for the detection of various types of hydrocephalus. However, the μs’ is quite sensitive to the scatter numbers and size, i.e., tissue composition and hemoglobin concentration may interfere with the measurement. Therefore, it may be interesting to study the profiles of μs’ from a larger population with all types of hydrocephalus, such as communicating, noncommunicating, and normal pressure hydrocephalus, or hydrocephalus caused by brain atrophy, traumatic injury, or psychiatric disorders, in future works.

## 4. Materials and Methods

### 4.1. Animal Preparation and Experimental Design

All animal experiments were approved by the Institutional Animal Care and Use Committee of the Laboratory Animal Center in National Cheng Kung University (IACUC Approval No.: 101203). Adult (8–10 weeks-old) male Sprague–Dawley (SD) rats weighting 290–330 g were obtained from the Animal Center of National Cheng Kung University Medical College. The animals were housed in standard cages at a temperature of 25 ± 1 °C with a 12/12-h light/dark cycle and free food and water access.

In total, 42 adult male SD rats were used in this study; 28 of them underwent MCAO surgery and 14 of them underwent sham surgery. NIRS assessments were performed at the following time courses: pre, 15 min before MCAO surgery was started; occ, 15 min after the MCA was occluded with insertion of the filament; post, 15 min after the filament was withdrawn from the MCA to allow reperfusion; 1d, 1 day; 2d, 2 days; 3d, 3 days after the surgery (Figure 5B). On the third day when NIRS measurement was finished, all the rats (42 rats) were then sacrificed for histological examination. Only slices from MCAO rats (14 + 14 = 28 rats) were quantified to obtain the infarction area (%) and swelling area (%).

### 4.2. MCAO Surgery

The MCAO model was established following Koizumi’s method [6]. Briefly, the rats were maintained anesthetized with 2–3% isoflurane inhalation (in 21% O_2_ mixed with 79% N_2_). Rectal temperature was monitored and kept at 37.0 °C using an electrical heating pad (TC-1000 Temperature Controller, CWE Inc., Ardmore, PA, USA). The rats were placed in the supine position with the neck shaved. A vertical incision was made on the neck. The left CCA, external carotid artery (ECA), and ICA were exposed and isolated from the surrounding nerves and fascia. The ECA and CCA were ligated with 6-0 silk sutures permanently. The bifurcation of the CCA into the ECA and ICA was identified, and an incision was made on the CCA. A silicone-coated filament (403945PK10Re, Doccol Corp., Sharon, MA, USA) was advanced into the ICA through the small incision on the CCA, until it reached the origin of the MCA (~19 mm beyond the bifurcation). The filament was secured with knots tightened around the ICA and CCA. Ninety minutes later, the filament was carefully removed from the ICA and CCA to allow reperfusion of the MCA, and the ICA was then permanently ligated at the bifurcation to prevent backflow through the arteriotomy. For the sham control, an animal was subjected to CCA, ECA, and ICA ligation without filament insertion. Following surgery, the incision on the neck was sutured and an intraperitoneal injection of 3.0 mL of saline was given as fluid replacement. To relieve pain after surgery, buprenorphine (0.1 mg/kg) was injected subcutaneously twice per day.

### 4.3. NIRS Measurement

The intrinsic cerebral optical properties, hemoglobin concentrations, tissue oxygen saturation, and inter-hemispheric hemodynamic oscillation were non-invasively monitored using a commercial FD-NIRS system (Imagent, ISS Inc., Champaign, IL, USA) [74,75,76]. The NIRS setup for the rat model involved 12 laser diodes at 690 and 830 nm (6 diodes for each wavelength) and 2 photomultipliers. The laser light at the 2 wavelengths was modulated at 110 MHz and coupled into 6 homemade source probes. Each source probe consisted of 2 optical fibers (core diameter = 400 μm; BFH22-200, Thorlabs Inc., Newton, NJ, USA) to guide laser beams at the 2 wavelengths into one location of the brain. The detector probe had 1 optical fiber connected to 1 photomultiplier. For each hemisphere, 3 source probes (2 wavelengths for each) were arranged at distances of 3.0, 6.0, and 9.0 mm, respectively, from the detector probe. Under anesthesia maintained with 2–3% isoflurane inhalation (in 21% O_2_ mixed with 79% N_2_), the rat was secured on a stereotaxic frame. The probes were held by a self-made holder and their tips were placed in good contact with the shaved scalp of the rat’s head. The probes were located at 12 mm beyond the interaural line on the midline (Figure 5A).

The instrument measured the amplitudes and phases of the frequency-modulated laser light at the two wavelengths and three source-detector separation distances for each hemisphere. Diffusion approximation, which is a simplified solution based on semi-infinite geometry for the photon diffusion equation, revealed linear relationships between the phases and logarithmic amplitudes and the multiple source–detector separation distances. By fitting the slopes of these linear relationships, the absolute quantitative measurement of μa and μs’ of brain tissues at each wavelength could be obtained [74]. Before each assessment, the system was calibrated using a silicone phantom with known μa and μs’ values. The absolute values of [HbO], [HbR], [HbT] ([HbT] = [HbO] + [HbR]), and StO_2_ (StO_2_ = ([HbO]/[HbT]) × 100%) were then calculated from the measured μa using the software, Boxy (Boxy, ISS Inc., Champaign, IL, USA). The representative datum from each time course was obtained by averaging the continuous recording for 60–100 s at a sampling rate of 5 Hz. IHCCs were obtained by calculating the zero-lag cross-correlation of the [HbO] traces from both hemispheres pre-filtrated with a 0.01–0.8 Hz band-pass filter using Matlab software (The MathWorks Inc., Natick, MA, USA).

### 4.4. Histological Examination

To estimate the extent of brain infarction induced by the MCAO surgery, coronal sections of the brain were stained with 2,3,5-triphenyltetrazolium chloride (TTC) (1.08380.0010, MilliporeSigma, Burlington, MA, USA) for histological analysis [77]. Briefly, the rats were deeply anesthetized with 5% isoflurane inhalation and perfused with normal saline. The cerebrum was extracted and then cut into eight coronal sections with 2-mm thickness. The sections were incubated in 1% TTC in normal saline for 30 min at room temperature. The sections were then washed in ice-cold saline and placed on a scanner (HP Scanjet 3600 series, HP Inc., Palo Alto, CA, USA) for imaging. After TTC staining, vital brain tissue appeared red and infarct areas appeared white. The percent of infarction was quantified following the previous protocol [78]. Image analysis was performed on gray-scale images, and the high and low thresholds were assigned to distinguish viable (red) and non-viable (white) tissues using ImageJ software (NIH, Bethesda, MD, USA). For consistent sets of thresholds across studies, consistent staining and scanning conditions are required. The volume of white tissue on the ipsilateral hemisphere (V_whi_), volume of the whole ipsilateral hemisphere (V_ips_), and volume of the whole contralateral hemisphere (V_con_) were determined using the measurement tool in ImageJ. To avoid overestimation caused by edema, the percent of infarction was calculated indirectly: (1)$$\mathrm{Infarction}\;\left(\%\right)=\frac{V_{\mathrm{con}}-V_{\mathrm{ips}}+V_{\mathrm{whi}}}{V_{\mathrm{con}}}\times 100\%$$

The brain swelling was estimated by calculating the percent of tissue swelling:(2)$$\mathrm{Swelling}\;\left(\%\right)=\frac{V_{\mathrm{ips}}-V_{\mathrm{con}}}{V_{\mathrm{con}}}\times 100\%$$

### 4.5. Data Analysis

Quantities of brain infarction from MCAO rats were partitioned into two clusters using the *k*-means algorithm in MATLAB (The MathWorks Inc., Natick, MA, USA) to determine the grouping of mild and severe infarction (Figure 2). To compare the degree of edema between groups, the two-tailed unpaired t-test was performed using the software GraphPad Prism (version 5.01; GraphPad Software, San Diego, CA, USA) with the significance set as *p* < 0.05. The percentages of edema are presented as means ± standard deviation (SD).

The mean value of each NIRS measurement was obtained by averaging the recording trace. The NIRS parameters were then grouped according to the size of the infarct revealed by TTC staining for further investigation of the relationship between hemodynamic changes and ischemic severity (Figure 3). Two-way repeated-measures ANOVA was performed to test whether there was any effect on these parameters contributed by the ischemic severity (sham, mild, and severe infarction), by the time courses of evolving ischemia, or by the interaction between time course and severity. When differences or interactions were detected, Tukey’s post-hoc test was applied to compare the means from sham, mild, and severe infarctions at various time courses. The data are presented as means ± standard error of the mean (SEM), and the significance level was set as *p* < 0.05. Since the sampling distribution of Pearson’s correlation coefficient is not normally distributed, Fisher’s Z-transform was applied to convert IHCCs to normally distributed variables. Two-way ANOVA and Tukey’s test were applied to the transformed IHCCs with the significance level set as *p* < 0.05. In the plots, all IHCC values are expressed as untransformed means ± SEM.

## 5. Conclusions

A reproducible and invariable preclinical animal study is critical for translational research on the treatment of stroke. Considering the nature of variabilities of ischemic infarction, it has become an urgent need to detect or early predict the outcome of animal models [20,22,27,29,50,53]. Featured as a real-time and non-invasive tool that can provide cerebral hemodynamics and optical properties, FD-NIRS has the potential to monitor the outcome of the MCAO surgery. In addition, the nature of absolute quantification makes FD-NIRS measurements the universal variables that are comparable among different batches or labs.

In the present study, we reported sequential changes relative to MCAO modeling, including the tissue optical properties, the cerebral hemodynamics, and the interhemispheric correlation of hemodynamic oscillation. Bimodal infarction severity was identified after MCAO surgery, based on which the FD-NIRS parameters were grouped and analyzed. When comparing the results from the severe infarction and sham groups, significant differences were shown in [HbO], [HbT], and StO_2_ during occlusion and d3, in IHCC after early reperfusion, and in tissue scattering on d2 and d3. However, there was no significant difference observed in most of the parameters derived from the mild infarction versus sham groups, except [HbO] and StO_2_ on d3. Therefore, our finding supports that FD-NIRS parameters may be associated with severe ischemia in the MCAO model. The potential application of FD-NIRS parameters as the indicators for success modeling requires further study to overcome the limitations mentioned earlier. Future clinical studies are also inspired to characterize the utility of NIRS parameters as early prognostic biomarkers [67] for ischemic stroke.

## Figures and Tables

**Figure 1 ijms-23-10318-f001:**
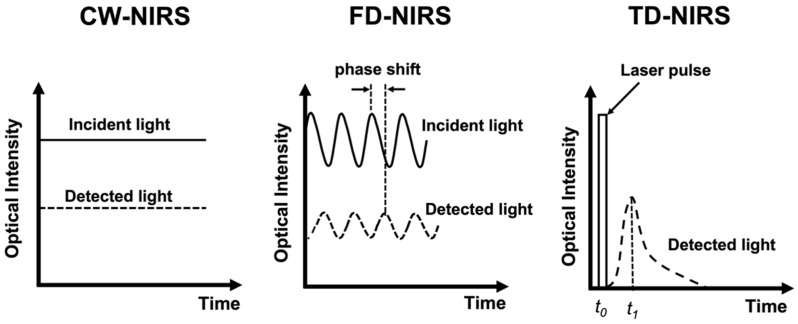
Schematic presentation of the principles of the CW-NIRS, FD-NIRS, and TD-NIRS.

**Figure 2 ijms-23-10318-f002:**
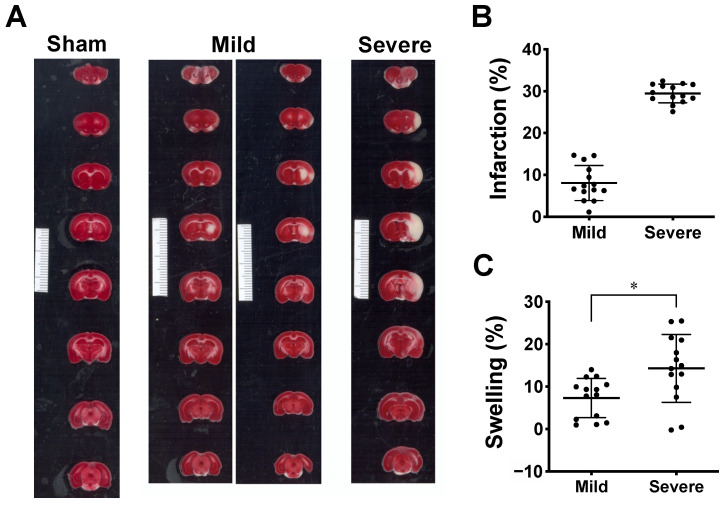
Ischemic lesion induced by MCAO model. (**A**) This shows 1% TTC staining of the brain slices collected on the third day after MCAO. The lesion appeared in white while the vital tissue was stained in red. The result shows two types of ischemic infarction: mild infarction (*n* = 14), rats with small-sized lesions mainly in subcortical tissue (left column of mild infarction), some involving small ventrolateral cortex (right column of mild infarction); severe infarction (*n* = 14), rats with large lesions covering major cortex and subcortical tissue. Minimum unit in scale bar: 1 mm. (**B**) Scatter plot showing distribution of infarct volume from two groups of MCAO rats partitioned using *k*-means clustering. Data are presented as means ± SD. (**C**). Tissue swelling of lesioned hemisphere from two groups was estimated from volumetric changes (%) between both hemispheres. Data are presented as means ± SD, * *p* < 0.05 using *t*-test.

**Figure 3 ijms-23-10318-f003:**
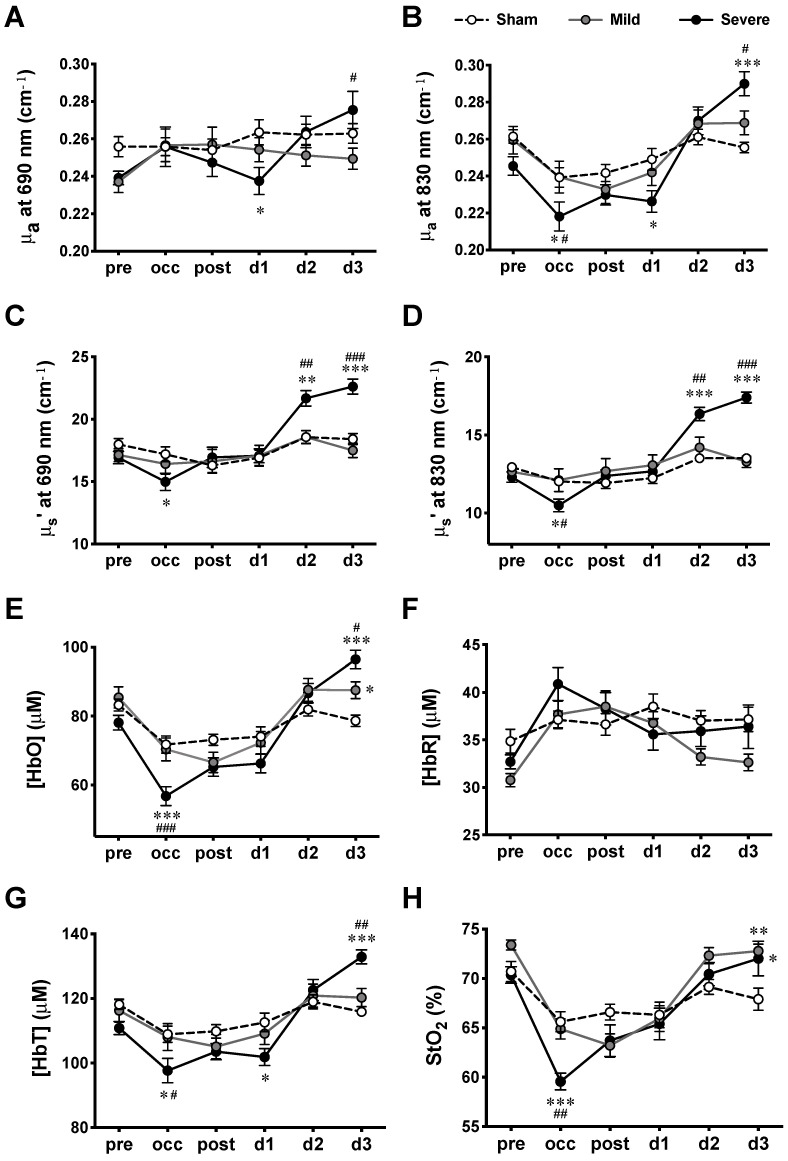
Hemodynamic measurements at various time courses in MCAO model. (**A**) μa at 690 nm wavelength, (**B**) μa at 830 nm wavelength, (**C**) μs’ at 690 nm wavelength, (**D**) μs’ at 830 nm wavelength, (**E**) [HbO], (**F**) [HbR], (**G**) [HbT], and (**H**). StO_2_ measured for sham, mild, and severe infarctions. Data are presented as means ± SEM. *n* = 14 for each group. *: *p* < 0.05, **: *p* < 0.01, ***: *p* < 0.001, versus sham group; #: *p* < 0.05, ##: *p* < 0.01, ###: *p* < 0.001, versus mild infarction.

**Figure 4 ijms-23-10318-f004:**
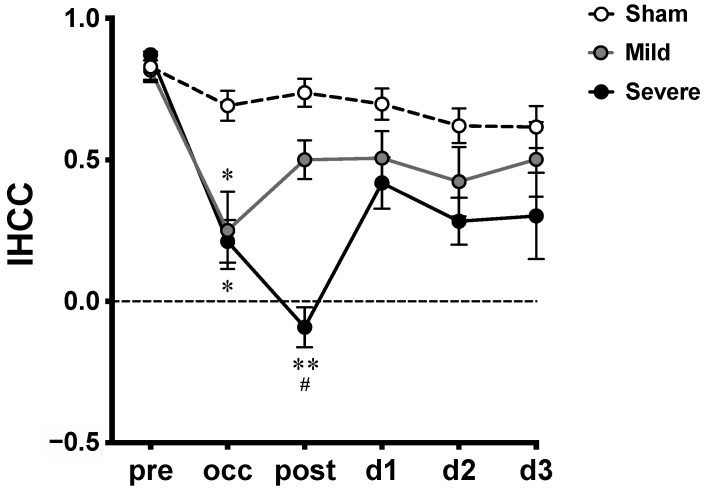
Interhemispheric correlation coefficients (IHCCs) measured at various time courses in MCAO model. Untransformed IHCCs calculated from [HbO] traces of both hemispheres are presented as means ± SEM; *: *p* < 0.01, **: *p* < 0.001, versus sham group; #: *p* < 0.01, versus mild infarction.

**Figure 5 ijms-23-10318-f005:**
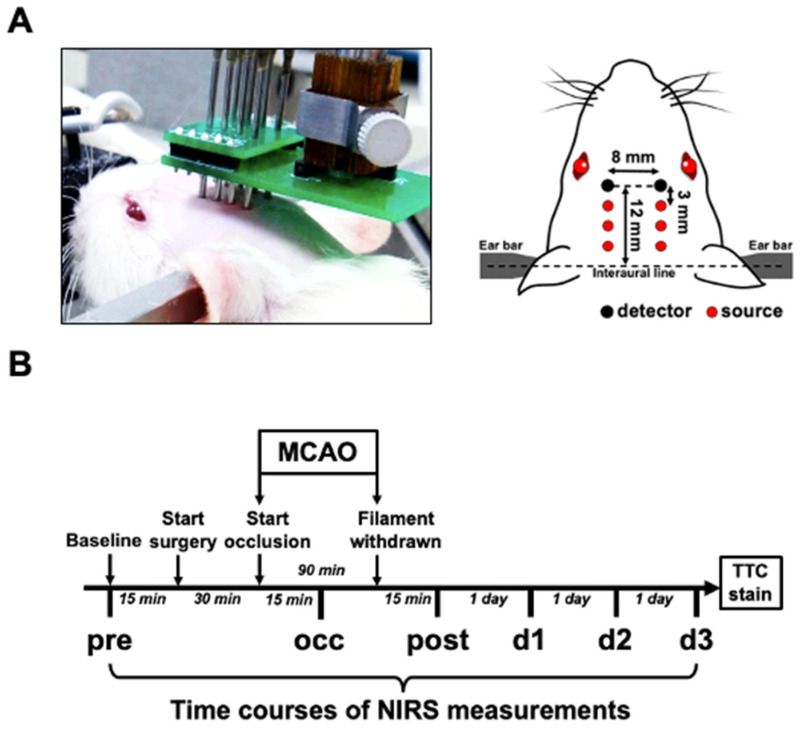
Experimental setup of bilateral FD-NIRS assessment of MCAO rat. (**A**) Schematic diagram showing optical probes arranged on rat’s head. Optical probes were positioned and firmly attached to the pre-shaved scalp. Each light source contains laser beams from 690 and 830 nm laser diodes. The relative positions of all the light sources and detectors are presented. (**B**) Scheme showing timeline of experimental design. FD-NIRS measurements were performed at the following time points: 15 min before surgery (pre), 15 min after MCAO was induced (occ), 15 min after reperfusion (post), and 1 day (d1), 2 days (d2), and 3 days (d3) after surgery.

**Table 1 ijms-23-10318-t001:** FD-NIRS measurements obtained at various time courses of MCAO rats.

Parameter	µ_a_ (690 nm) (cm^−1^)			µ_a_ (830 nm) (cm^−1^)			µ_s_’ (690 nm) (cm^−1^)			µ_s_’ (830 nm) (cm^−1^)		
Group	Sham	Mild	Severe	Sham	Mild	Severe	Sham	Mild	Severe	Sham	Mild	Severe
pre	0.256 ± 0.005	0.237 ± 0.006	0.239 ± 0.004	0.261 ± 0.004	0.260 ± 0.007	0.245 ± 0.005	17.98 ± 0.48	17.75 ± 0.52	16.93 ± 0.50	12.94 ± 0.25	12.64 ± 0.37	12.30 ± 0.33
occ	0.256 ± 0.005	0.257 ± 0.009	0.256 ± 0.011	0.240 ± 0.005	0.240 ± 0.009	0.218 ± 0.008	17.20 ± 0.60	16.42 ± 0.83	14.99 ± 0.69	12.03 ± 0.26	12.11 ± 0.73	10.49 ± 0.41
post	0.254 ± 0.006	0.257 ± 0.009	0.247 ± 0.007	0.242 ± 0.005	0.233 ± 0.008	0.230 ± 0.006	16.30 ± 0.56	16.64 ± 0.96	16.95 ± 0.82	11.93 ± 0.36	12.68 ± 0.81	12.38 ± 0.38
d1	0.264 ± 0.007	0.254 ± 0.006	0.238 ± 0.007	0.249 ± 0.006	0.242 ± 0.007	0.226 ± 0.006	16.92 ± 0.62	17.08 ± 0.84	17.08 ± 0.56	12.24 ± 0.34	13.07 ± 0.66	12.67 ± 0.28
d2	0.262 ± 0.006	0.251 ± 0.006	0.264 ± 0.008	0.261 ± 0.004	0.268 ± 0.007	0.270 ± 0.008	18.57 ± 0.53	18.58 ± 0.51	21.68 ± 0.62	13.51 ± 0.24	14.19 ± 0.66	16.35 ± 0.43
d3	0.263 ± 0.005	0.249 ± 0.006	0.275 ± 0.010	0.255 ± 0.003	0.269 ± 0.006	0.290 ± 0.007	18.41 ± 0.44	17.51 ± 0.57	22.62 ± 0.60	13.51 ± 0.12	13.29 ± 0.36	17.40 ± 0.35
Parameter	[HbO] (µM)			[HbR] (µM)			[HbT] (µM)			StO_2_ (%)		
Group	Sham	Mild	Severe	Sham	Mild	Severe	Sham	Mild	Severe	Sham	Mild	Severe
pre	83.25 ± 1.73	85.48 ± 3.06	78.13 ± 2.14	34.86 ± 1.25	30.77 ± 0.69	32.70 ± 0.73	118.1 ± 1.5	116.3 ± 3.6	110.8 ± 2.0	70.70 ± 1.03	73.40 ± 0.52	70.35 ± 0.84
occ	71.78 ± 2.51	70.36 ± 3.33	56.79 ± 2.74	37.12 ± 0.93	37.69 ± 1.40	40.88 ± 1.72	108.9 ± 2.5	108.1 ± 4.2	97.7 ± 3.8	65.62 ± 1.02	64.89 ± 1.03	59.56 ± 0.86
post	73.15 ± 1.63	66.57 ± 2.97	65.23 ± 2.70	36.64 ± 1.14	38.50 ± 1.67	38.28 ± 1.71	109.8 ± 2.1	105.1 ± 3.8	103.5 ± 2.5	66.58 ± 0.81	63.21 ± 1.20	63.71 ± 1.61
d1	74.09 ± 2.86	72.33 ± 3.28	66.28 ± 2.73	38.50 ± 1.34	36.77 ± 1.21	35.58 ± 1.66	112.6 ± 2.9	109.1 ± 3.4	101.9 ± 2.6	66.31 ± 1.23	65.93 ± 1.31	65.41 ± 1.60
d2	81.96 ± 1.87	87.67 ± 3.30	86.69 ± 2.85	37.03 ± 1.06	33.21 ± 0.85	35.93 ± 1.62	119.0 ± 2.2	120.9 ± 3.6	122.6 ± 3.3	69.14 ± 0.75	72.32 ± 0.82	70.44 ± 1.18
d3	78.67 ± 1.65	87.58 ± 2.46	96.47 ± 2.63	37.16 ± 1.28	32.64 ± 0.88	36.41 ± 2.29	115.8 ± 1.3	120.3 ± 2.8	132.9 ± 2.2	67.90 ± 1.11	72.78 ± 0.70	72.02 ± 1.75

*n* = 14 for each group. Data are presented as means ± SEM.

**Table 2 ijms-23-10318-t002:** Summarized F and *p*-values derived from two-way repeated-measure ANOVA of effects of factors (time course, severity of infarction, and interaction between them) on cerebral optical properties (µa and µs’) and hemodynamics parameters ([HbO], [HbR], [HbT], and StO_2_).

Factor	µ_a_ (690 nm)	µ_a_ (830 nm)	µ_s_’ (690 nm)	µ_s_’ (830 nm)
Time course	F_5,195_ = 4.35, *p* = 0.0009	F_5,195_ = 41.29, *p* < 0.0001	F_5,195_ = 28.71, *p* < 0.0001	F_5,195_ = 41.53, *p* < 0.0001
Severity	F_2,39_ = 0.69, *p* = 0.5063	F_2,39_ = 0.38, *p* = 0.6873	F_2,39_ = 1.57, *p* = 0.2201	F_2,39_ = 2.24, *p* = 0.1201
Interaction	F_10,195_ = 2.87, *p* = 0.0023	F_10,195_ = 6.55, *p* < 0.0001	F_10,195_ = 9.01, *p* < 0.0001	F_10,195_ = 12.05, *p* < 0.0001
Factor	[HbO]	[HbR]	[HbT]	StO_2_
Time course	F_5,195_ = 72.10, *p* < 0.0001	F_5,195_ = 11.59, *p* < 0.0001	F_5,195_ = 41.29, *p* < 0.0001	F_5,195_ = 51.30, *p* < 0.0001
Severity	F_2,39_ = 0.76, *p* = 0.4744	F_2,39_ = 1.22, *p* = 0.3068	F_2,39_ = 0.34, *p* = 0.7138	F_2,39_ = 1.35, *p* = 0.2717
Interaction	F_10,195_ = 10.33, *p* < 0.0001	F_10,195_ = 2.56, *p* = 0.0062	F_10,195_ = 7.48, *p* < 0.0001	F_10,195_ = 5.63, *p* < 0.0001

## Data Availability

Not applicable.

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
