# Peer review of "Hemodynamics and Tissue Optical Properties in Bimodal Infarctions Induced by Middle Cerebral Artery Occlusion"

_ijms, 2022, doi:10.3390/ijms231810318_

Round 1
Reviewer 1 Report
Present study utilized the FD-NIRS method to determine the relationship between cerebral hemodynamics and infarct variation in middle cerebral artery occlusion (MCAO)-rat model. Using intraluminal filament for performing MCAO in rats, authors observed significant decreased in cerebral hemodynamics and optical properties in animals with severe infarction. Authors have performed interhemispheric correlation coefficient (IHCC) from low frequency hemodynamics oscillation in both hemispheres. Authors have observed lower IHCCs standing for interhemispheric desynchronization in both mild and severe infarction during occlusion, and only in severe infarction after reperfusion. Authors suggested that the sequential FD-NIRS parameters may be associated with the severity of the infarction in MCAO model, as well as with the pathologies such as vascular dysfunction and brain edema.
The aim of the present study is pretty straight forward and easy to follow. Authors have assessed the experimental brain infarctions made through MCAO and applied FD-NIRS technique to determine the brain damage. The present manuscript is written reasonably well and have followed the relevant literature in the present field. I do not have major comments to obstruct/or delay in accepting this manuscript. However, there are some minor corrections that needs to be amended in the manuscript before it’s good to go! Please see below these minor comments and respond accordingly
Good luck.
Abstract – In the abstract please specify animal model.
Introduction – Introduction need to be concise with information relevant to the subject of this study. Please shorten the introduction with only couple of paragraphs.
Figure 1, should not appear in the introduction section before the results. This will confuse the readers.
Materials and Methods:
4.1 Animal Preparation:
Line 444, - “weighting” – typos!
Discussion:
3.1 MCAO variation and LDF Monitoring
I think authors forget to remove the pervious reviewer comments which appeared in the discussion section under 3.1 section. Please remove those comments.
Author Response
Response to Reviewer 1 Comments
Point 1: Abstract – In the abstract please specify animal model.
Response 1: We thank reviewer for the correction. The following sentences have been added in Abstract: “……in MCAO model. The rats were undergone transient MCAO using intraluminal filament. The optical properties and hemodynamics were measured by placing the FD-NIRS probes on the scalp of the head before, during and various time-courses after MCAO. Bimodal infarction severities were observed after the same 90-min MCAO condition using intraluminal filament.”(page 1)
Point 2: Introduction – Introduction need to be concise with information relevant to the subject of this study. Please shorten the introduction with only couple of paragraphs.
Response 2: We thank reviewer for the comment. The first and second paragraphs in Introduction have been trimmed as following (page 2). And the remain contents have been merged into one paragraph:
Original 1st paragraph: “……The middle cerebral artery (MCA) is the largest cerebral artery that supplies most of the brain cerebrum, including parts of the frontal, temporal, and parietal lobes, as well as the basal ganglia and thalamus [4]. The MCA and is also the cerebral artery most often affected by stroke: about half (50.8%) of all ischemic strokes occur in MCA and its branches [4,5]. For this reason, middle cerebral artery occlusion (MCAO) using intraluminal filament to generate infarcts in MCA territory has been developed in 1980s [6,7]. The model involves introducing a nylon filament directly into the common carotid artery (CCA) or external carotid artery (ECA), and then advancing the filament through internal carotid artery (ICA) until it blocks the cerebral blood flow (CBF) to the MCA [3]. The tip of the nylon filament can be rounded by heating near a flame [7], or be coated with silicone [6] or poly-L-lysine [8] to provide superior blockage. This method does not require craniectomy and allows permanent occlusion or temporary one followed by reperfusion.”
Original 2nd paragraph: “Parameters that can affect the infarct variability and outcome reproducibility of MCAO model includes strains, age, body weight, body temperature, anesthesia, occlusion duration, type of intraluminal filament and other surgical procedures [13]. These parameters are usually decided according to the follow-up intervention and endpoint analysis so the aim of the animal study can be achieved. Despite many studies lead optimization of the methodological parameters [13], the intraluminal placement of the filament is still a surgical challenge…….These disadvantages limit the use of LDF, LSCI and MRI for during MCAO surgery monitoring. A better way to monitor the cerebral blood supply during MCAO for researcher to confirm the success of the modeling, and to explore the pathological changes after MCAO is still a significant demand.”
Point 3: Figure 1 should not appear in the introduction section before the results. This will confuse the readers.
Response 3: We thank reviewer for the correction. The Figure 1 has been moved to section of Results.
Point 4: Materials and Methods – 4.1 Animal Preparation: Line 444, - “weighting” – typos!
Response 4: We thank reviewer for the correction. The following sentence in 4.1 Animal Preparation has been corrected: “……Adult (8–10 weeks-old) male SD rats weighting with weight of 290–330 g were obtained from……” (page 15)
Point 5: Discussion – 3.1 MCAO variation and LDF Monitoring: I think authors forget to remove the pervious reviewer comments which appeared in the discussion section under 3.1 section. Please remove those comments.
Response 5: We thank reviewer for the correction. The content has been removed. (page 10)

Reviewer 2 Report
Dear Editor,
Hemodynamics and Tissue Optical Properties in Bimodal Infarctions Induced is an interesting topic for the readers of the Journal. Specifically, the method that the author used for noninvasive analysis of the hemodynamics and tissue optical properties was really interesting for me. The method is well-described. Some parts of the Result section were repetitive, but in total, the results and discussion section also support the hypothesis and main claim of the author in the last paragraph of the Introduction section. There are some notes for better using these results for clinicians and for direct clinical applications that are useful to increase the quality o this paper. I suggest publishing the paper after considering the following comments.
1. “Significant decreases in cerebral hemodynamics and optical properties were detected during the occlusion in the 31 severe infarction but not the mild one”. What does “decreases in hemodynamics” mean? Also, increase in hemodynamics in the next sentence. Hemodynamic parameters is a very general phrase that includes many parameters like velocity, pressure, etc. in the main text you described your intention about this phrase but this general phrase in the Abstract does not make sense.
2. The abstract does not include the “method” of this study.
3. There are various models for the hydrodynamic analysis of blood and intracranial fluid characteristics such as computer simulation [https://doi.org/10.1134/S0021894417060025], [https://doi.org/10.1016/j.wneu.2019.11.171], and [https://doi.org/10.1007/s10143-020-01367-3]. Please make a big picture in the second paragraph of the introduction for the readers using all existing methods, then focus on FD-NIRS.
4. “…MCAO model includes strains, a…”. Can you please explain more about the effect of “Strain” on the “MCAO model”.
5. Do you have any idea about the possible dependency of the parameters like “[HbO], [HbR], and [HbT]” (or other parameters), and consequently, using Bonferonni corrected p-value instead of 0.05? If, yes, what will be your conclusion about parameters with “*” and “#” in Fig. 2.
6. The main part of Lines 163-172 can be a part of the Method, not the result section. Or move to 4.6 or move to the statistical analysis section.
7. Can you please explain more about how to distinguish the groups: Mild and Severe?
8. Section”3.2” can be the first section of the Discussion.
9. In some places such as the first paragraph of section 3.3, you repeat a part of your results, but you did not make at least a primary discussion on this and somehow you give up your findings.
10. The findings are really valuable and interesting. For practical use of this achievement, can you please add some sentences for direct connecting of your results for clinical applications for using the clinicians? You discussed “Delayed increases in µs’” or “brain swelling induced by water intoxication”. For example, can you connect “brain swelling induced by water intoxication” to cerebral atrophy or any other CNS disorders? MCA has a high potential for an aneurysm. Do you think your “Hemodynamic” analysis can be useful for this? Brain compliance is a very effective clinical parameter to study “brain edema” and this has a considerable impact on swelling and brain properties [https://doi.org/10.3389/fbioe.2022.900644].
11. Can you please add the sample size/number of animals and even slices in the Method section: Animal Preparation section?
12. Do you have any strong preference for this concern: translating your animal findings to human subjects?
Author Response
Response to Reviewer 2 Comments
Point 1: “Significant decreases in cerebral hemodynamics and optical properties were detected during the occlusion in the severe infarction but not the mild one”. What does “decreases in hemodynamics” mean? Also, increase in hemodynamics in the next sentence. Hemodynamic parameters is a very general phrase that includes many parameters like velocity, pressure, etc. in the main text you described your intention about this phrase but this general phrase in the Abstract does not make sense.
Response 1: We thank reviewer for the comment. To describe the finding in a more specific way, we have modified the following sentences in Abstract: “……Significant decreases in cerebral hemodynamics and optical properties concentrations of oxygenated hemoglobin ([HbO]) and total hemoglobin ([HbT]), tissue oxygenation saturation (StO2), absorption coefficient (ma) at 830 nm, and reduced scattering coefficient (ms’) at both 690 nm and 830 nm were detected during the occlusion in the severe infarction but not the mild one. Of note, the significant increases in scattering and hemodynamics [HbO], [HbT], StO2, and ma at both 690 nm and 830 nm were found on day 3; and increases in ms’ at both 690 nm and 830 nm were found on day 2 and day 3 after MCAO, respectively.” (page 1)
Point 2: The abstract does not include the “method” of this study.
Response 2: We thank reviewer for the correction. The following sentences have been added in Abstract: “……in MCAO model. The rats were undergone transient MCAO using intraluminal filament. The optical properties and hemodynamics were measured by placing the FD-NIRS probes on the scalp of the head before, during and various time-courses after MCAO. Bimodal infarction severities were observed after the same 90-min MCAO condition using intraluminal filament.” (page 1)
Point 3: There are various models for the hydrodynamic analysis of blood and intracranial fluid characteristics such as computer simulation [https://doi.org/10.1134/S0021894417060025], [https://doi.org/10.1016/j.wneu.2019.11.171], and [https://doi.org/10.1007/s10143-020-01367-3]. Please make a big picture in the second paragraph of the introduction for the readers using all existing methods, then focus on FD-NIRS.
Response 3: We thank reviewer for the comment. (1) The hydrodynamic analysis of blood and intracranial fluid characteristics using computer simulation has been discussed in new added section 3.6, second paragraph (page 15): “The current finding is made from ischemic stroke model, while the hemorrhagic stroke which contributes 20% of the total stroke is also critical [1,2]. Intracranial aneurysm (IA) is a risk for hemorrhagic stroke, and MCA has a high potential for developing an IA [68]. The relationship between changes in CBF and wall shear stress have been studied for the rupture risk using computer simulation [69,70]. Meanwhile, cerebral autoregulation has been showed disrupted after aneurysmal subarachnoid hemorrhage [71,72]. Therefore, we speculate that IHCC, which is affected by disrupted cerebral autoregulation [37], may be a potential indicator for aneurysmal subarachnoid hemorrhage detection.” (2) And a picture summarizing the NIRS method has been added to Introduction, new Figure 1.
New added references cited in this paragraph:
- Gholampour, S.; Mehrjoo, S. Effect of bifurcation in the hemodynamic changes and rupture risk of small intracranial aneurysm. Neurosurgical Review 2021, 44, 1703-1712, doi:10.1007/s10143-020-01367-3.
- Hajirayat, K.; Gholampour, S.; Sharifi, I.; Bizari, D. Biomechanical Simulation to Compare the Blood Hemodynamics and Cerebral Aneurysm Rupture Risk in Patients with Different Aneurysm Necks. Journal of Applied Mechanics and Technical Physics 2017, 58, 968-974, doi:10.1134/S0021894417060025.
- Taher, M.; Gholampour, S. Effect of Ambient Temperature Changes on Blood Flow in Anterior Cerebral Artery of Patients with Skull Prosthesis. World Neurosurgery 2020, 135, e358-e365, doi:https://doi.org/10.1016/j.wneu.2019.11.171.
- Lidington, D.; Wan, H.; Bolz, S.S. Cerebral Autoregulation in Subarachnoid Hemorrhage. Front Neurol 2021, 12, 688362, doi:10.3389/fneur.2021.688362.
- Lang, E.W.; Diehl, R.R.; Mehdorn, H.M. Cerebral autoregulation testing after aneurysmal subarachnoid hemorrhage: the phase relationship between arterial blood pressure and cerebral blood flow velocity. Crit Care Med 2001, 29, 158-163, doi:10.1097/00003246-200101000-00031.
Point 4: “…MCAO model includes strains, a…”. Can you please explain more about the effect of “Strain” on the “MCAO model”.
Response 4: We thank reviewer for the comment. To better elaborate the effect of rat strain on the MCAO model, we have added the following content in 3. Discussion, 3.5 Limitations of the Study, second paragraph: (page 14)“Furthermore, the anatomical variations in cerebrovasculature associated with strains can affect variation of infarction [13,14,55]. A meta-analysis compared the coefficient of variation of infarct size induced by various MCAO protocols in different rat strains, Wistar has the significant lower variability than Sprague-Dawley (SD) [13]. Variance in infarct size from SD rats may be due to the ICA branches that can support MCA territory were identified in 80% of the animals [14]. A review article then discussed that Wistar Kyoto (WKY) is a better choice than SD for stroke modeling because it lacks the vascular variability of the SD [55]. Theoretically, vascular variation shall affect NIRS result as well. Parallel study using brain imaging to provide structural information and spatial hemodynamics is critical for validation.”
Point 5: Do you have any idea about the possible dependency of the parameters like “[HbO], [HbR], and [HbT]” (or other parameters), and consequently, using Bonferonni corrected p-value instead of 0.05? If, yes, what will be your conclusion about parameters with “*” and “#” in Fig. 2.
Response 5: We thank reviewer for the comment. The [HbO] and [HbR] are derived from ma and ms’ at both 690 and 830 nm. The [HbT] is the sum of the [HbO] and [HbR]. Therefore, the sequential patterns of the [HbO], [HbR] and [HbT] are showing some dependency. And if we consider using Bonferonni correction to control type I error, i.e., deviding the significant level alpha by 3: new alpha = 0.05/3 = 0.0167 (we are comparing any possible pairs among sham, mild and severe groups at each time point, so there are 3 comparison: mild v.s. sham, severe v.s. sham, severe v.s. mild at each time point), in this case, all the comparisons between any paired parameters with significance level of “*” and “#” in Figure 2 will become non-significant. However, in the current study we were using Tuckey test for post-hoc multiple comparison for the following reseasons: 1. Tuckey test has medium conservativeness while the Bonferonni is much rigorous [1]; 2. Tuckey test is based on the same sample counts [1] between groups (we have not a few sample size N = 14 for all the groups). Therefore, we were using Tuckey test to prevent type II error.
- Lee S, Lee DK. What is the proper way to apply the multiple comparison test? Korean J Anesthesiol. 2018 Oct;71(5):353-360. doi: 10.4097/kja.d.18.00242.
Point 6: The main part of Lines 163-172 can be a part of the Method, not the result section. Or move to 4.6 or move to the statistical analysis section.
Response 6: We thank reviewer for the suggestion. The contents in 2.2 have been moved to 4.6 Data Analysis, second paragraph (page 18): “Mean value of each NIRS measurement was obtained by averaging the recording trace. The NIRS parameters were then grouped according to the size of the infarct revealed by TTC staining for further investigation of the relationship between hemodynamic changes and ischemic severity (Figure 2). Two-way repeated-measures ANOVA was performed to test whether there was any effect on these parameters contributed by the ischemic severity (sham, mild and severe infarction), by the time courses of evolving ischemia, or by the interaction between time course and severity. Two-way repeated-measures ANOVA was used to determine whether the time course or infarct severity affected NIRS measurements, or whether there was any interaction between them. When differences or interactions were detected, the Tukey’s post-hoc test was applied……”
Point 7: Can you please explain more about how to distinguish the groups: Mild and Severe?
Response 7: We thank reviewer for the comment. To better explain the observation of mild and severe groups, we have modified the content in Result section, 2.1 MCAO Lesion (pages 4-5): “Although the consistent MCAO surgery was performed, TTC staining on the third day after MCAO showed two types of outcomes: (i) small-sized infarctions mainly in subcortical tissue, some involving the ventrolateral cortex, and (ii) large infarctions covering subcortical tissue and the ventrolateral and dorsolateral cortices (white signals in Figure 1A). These two types of infarction sizes were confirmed by clustering analysis using k-means algorithm. Based on the clustering, the outcomes of the infarction MCAO rats can be divided into the mild infarction group and the severe infarction group for further analysis (Figure 1A and B). The percent of infarction was 8.08 ± 4.19% (mean ± SD) for mild infarction (N = 14) and 29.44 ± 2.23% for severe infarction (N = 14). The NIRS parameters derived from mild and severe groups were than further analyzed for statistical difference (see section 2.2)…….”
Point 8: Section”3.2” can be the first section of the Discussion.
Response 8: We thank reviewer for the suggestion. The section 3.2 have been moved to 3.1.
Point 9: In some places such as the first paragraph of section 3.3, you repeat a part of your results, but you did not make at least a primary discussion on this and somehow you give up your findings.
Response 9: We thank reviewer for the comment. The redundant content in the first paragraph of discussion section 3.3 has been removed. And the paragraph has been re-written as followed (page 11): “In the current study, the severe infarction group expressed decreased ms’ during MCAO, and increased ms’ on d2 and d3 after MCAO. the values of ms’ from severe infarction at both 690 and 830 nm decreased during MCAO comparing to the values from sham group. Then the values increased on d2 and d3 when comparing with those from mild and sham groups. These sequential pattern of ms’ is not like simultaneously recorded ma, hemoglobin level or StO2 which are much relative to the changes in concentration of hemoglobin [32]. We assume the decrease in ms’ during MCAO may be due to the changes in cellular conformation induced by acute injury, and the increase in ms’ on d2 and d3 may be associated with brain edema induced by ischemic injury hours to days later. Therefore, to understand the physiological signification of our To further interpret our assumptions on the observations in ms’, several studies with consistent or controversial findings are discussed in the following paragraphs.”
Point 10: The findings are really valuable and interesting. For practical use of this achievement, can you please add some sentences for direct connecting of your results for clinical applications for using the clinicians? You discussed “Delayed increases in µs’” or “brain swelling induced by water intoxication”. For example, can you connect “brain swelling induced by water intoxication” to cerebral atrophy or any other CNS disorders? MCA has a high potential for an aneurysm. Do you think your “Hemodynamic” analysis can be useful for this? Brain compliance is a very effective clinical parameter to study “brain edema” and this has a considerable impact on swelling and brain properties [https://doi.org/10.3389/fbioe.2022.900644].
Response 10: We thank reviewer for the comment. The relationship between µs’ and its potential use for hydrocephalus has been discussed in new added section 3.6, third paragraph (page 15): “Non-invasive monitoring of the intracranial fluid content, such as intracranial compliance using MRI, has become an emerging need for clinical use in various types of hydrocephalus [73]. In the previous paragraph (section 3.3), we have discussed the increased ms’ may be associated with brain edema [59,60]. Since the ms’ obtained by non-invasive NIRS is an absolute quantity, ms’ is comparable between individuals and may have the potential for detection of various types of hydrocephalus. However, the ms’ is quite sensitive to the scatter numbers and size, i.e., tissue composition and hemoglobin concentration may interfere with the measurement. Therefore, it may be interesting to study the profiles of ms’ from larger population with all types of hydrocephalus, such as communicating, noncommunicating and normal pressure hydrocephalus, or hydrocephalus caused by brain atrophy, traumatic injury or psychiatric disorders in future works.”
New added references cited in this paragraph:
- Gholampour, S.; Yamini, B.; Droessler, J.; Frim, D. A New Definition for Intracranial Compliance to Evaluate Adult Hydrocephalus After Shunting. Frontiers in Bioengineering and Biotechnology 2022, 10, doi:10.3389/fbioe.2022.900644.
Point 11: Can you please add the sample size/number of animals and even slices in the Method section: Animal Preparation section?
Response 11: We thank reviewer for the comment. To describe the sample size in a more specific way, we have moved the section 4.4 Experimental Design to section 4.1 Animal Preparation, second paragraph (page 15): “Forty-two adult male SD rats were used in this study; 28 of them were underwent MCAO surgery and 14 of them underwent sham surgery. NIRS assessments were performed at the following time courses: pre, 15 minutes before MCAO surgery was started; occ, 15 minutes after the MCA was occluded with insertion of the filament; post, 15 minutes after the filament was withdrawn from the MCA to allow reperfusion; 1d, 1 day; 2d, 2 days; 3d, 3 days after the surgery (Figure 4B). On the third day when NIRS measurement was finished, all the rats (42 rats) were then sacrificed for histological examination. Only slices from MCAO rats (14+14 = 28 rats) were quantified to obtain the infarction area (%) and swelling area (%).”
Point 12: Do you have any strong preference for this concern: translating your animal findings to human subjects?
Response 12: We thank reviewer for the comment. We agree that translation of the NIRS study to clinical study is significant and we have added a new discussion section 3.6, first paragraph (pages 14-15): “Continuous measuring hemoglobin and tissue oxygenation level using NIRS may benefit acute stoke patients [32-34]. In the current study, we identified sequential changes of the brain hemodynamics and optical properties acquired with NIRS during and after rat MCAO. These findings may inspire the applications of NIRS for early diagnosis of ischemic stroke since NIRS is quite safe, affordable and feasible for long-term use [67]. However, the translational potential of these parameters in human stroke requires further studies because of the complexity of the stroke onset, individual variation of the stroke patient, and reproducibility of the NIRS measurement.”

Round 2
Reviewer 2 Report
Accept